# The Laws of Attraction: Chemokines as Critical Mediators in Cancer Progression and Immunotherapy Response in Bladder Cancer

**DOI:** 10.3390/cancers16193303

**Published:** 2024-09-27

**Authors:** Zaineb Hassouneh, Michelle E. Kim, Natalia Bowman, Manjeet Rao, Nu Zhang, Gang Huang, Robert S. Svatek, Neelam Mukherjee

**Affiliations:** 1Department of Urology, University of Texas Health Science Center at San Antonio (UTHSCSA), San Antonio, TX 78229, USA; hassouneh@livemail.uthscsa.edu (Z.H.); kime6@livemail.uthscsa.edu (M.E.K.); bowmann@livemail.uthscsa.edu (N.B.); svatek@uthscsa.edu (R.S.S.); 2Department of Microbiology, Immunology, and Molecular Genetics, University of Texas Health Science Center at San Antonio (UTHSCSA), San Antonio, TX 78229, USA; zhangn3@uthscsa.edu; 3Department of Cell Systems and Anatomy, Greehey Children’s Cancer Research Institute, San Antonio, TX 78229, USA; raom@uthscsa.edu (M.R.); huangg1@uthscsa.edu (G.H.)

**Keywords:** chemokines, migration, BCG, immune therapy, bladder cancer

## Abstract

**Simple Summary:**

Immunotherapeutic treatments have shown promising results in bladder cancer; however, patient responses remain low, often due to insufficient immune cell infiltration. Chemotactic cytokines, or chemokines, play a crucial role in mediating immune cell trafficking and activation, modulating the balance between immunoregulation and inflammation. In cancer, many chemokines are recognized for their skewed function, promoting the infiltration of immunosuppressive cells. Additionally, immunoregulatory chemokines contribute to tumor progression and metastasis by activating signaling pathways that upregulate angiogenesis and epithelial-to-mesenchymal transition. Characterizing the role of chemokines in bladder cancer is essential for identifying novel therapeutic targets and enhancing intratumoral immune infiltration.

**Abstract:**

Bladder cancer (BCa) is a prevalent urogenital malignancy, characterized by a myriad of genetic and environmental risk factors that drive its progression. Approximately 75% of bladder tumors are non-muscle-invasive at diagnosis. For such cases, bladder preservation is often feasible with intravesical chemotherapy or immunotherapy. However, the high recurrence rates associated with these tumors necessitate multiple cystoscopic examinations and biopsies, leading to significant financial burden and morbidity. Despite bladder tumors exhibiting one of the highest cancer mutational loads, which typically correlates with improved responses to immunotherapy, challenges persist. The tumor microenvironment serves as a nexus for interactions between tumor cells and the immune system, wherein chemokines and chemokine receptors orchestrate the recruitment of immune cells. This review addresses existing gaps in our understanding of chemokine dynamics in BCa by elucidating the specific roles of key chemokines in shaping the immune landscape of the tumor microenvironment (TME). We explore how dysregulation of chemokine signaling pathways contributes to the recruitment of immunosuppressive cell populations, such as Tregs and monocytes, leading to an unfavorable immune response. Additionally, we highlight the potential of these chemokines as predictive biomarkers for tumor progression and treatment outcomes, emphasizing their role in informing personalized immunotherapeutic strategies. By integrating insights into chemokine networks and their implications for immune cell dynamics, this review seeks to provide a comprehensive understanding of the interplay between chemokines and the immune microenvironment in BCa. Furthermore, we discuss the potential of targeting these chemokine pathways as innovative immunotherapeutic strategies, paving the way for enhanced treatment responses and improved patient outcomes.

## 1. Background

Bladder cancer (BCa) is the most common urinary tract malignancy and the fourth most common cancer in men with over 500,000 new cases and 200,000 deaths nationally [1,2]. Non-muscle-invasive BCa (NMIBC) is diagnosed in 70–75% of patients, with the remaining 25–30% of cases classified as muscle-invasive BCa (MIBC) [2]. While the mortality rate of low-grade NMIBC ranges from 3.76 to 10.88%, it increases dramatically to 19.52% in high-grade NMIBC [3]. The risk of recurrence and progression in high-grade NMIBC also increases by 45% and 17%, respectively [4]. The current gold standard of NMIBC treatment is transurethral resection of the bladder tumor (TURBT) followed by intravesical instillation of bacillus Calmette Guérin (BCG) [1]. However, 30% to 40% of patients experience BCG failure, making novel, alternative treatments crucial [5]. 

BCa is a highly immunogenic malignancy due to its high tumor mutational burden, making it a favorable target for immunotherapies such as immune checkpoint inhibitors (ICIs) [2]. Despite this, the response rate to ICIs remains low at 20–30%, which can be attributed to the immunosuppressive tumor microenvironment (TME) and the lack of immune infiltration [6,7]. Infiltration of cytotoxic immune cells into the tumor is modulated by the chemotactic activity of cytokines present in the TME, specifically chemokines [8,9]. In malignancies, the recruitment of inflammatory and cytotoxic immune cells is crucial. However, many pro-inflammatory chemokines display altered mechanisms and, in turn, contribute to a pro-tumorigenic immune environment, increasing tumorigenesis, angiogenesis, and metastasis [9]. 

## 2. Chemokine Family and Their Classifications

Chemotactic cytokines, known as chemokines, are small, highly conserved proteins that represent the largest family within cytokines [10]. A standardized nomenclature for chemokines was established in 2000, classifying them based on the location of cysteine residues at the N-terminus [11]. These cysteine residues are crucial for maintaining the structural integrity of chemokines, forming disulfide bonds that result in a secondary protein structure consisting of a triple-stranded β-sheet and an α-helix [11]. Based on these cysteine residues, chemokines are divided into four subgroups: CC, CXC, CX_3_C, and XC, with “L” denoting ligands and numbered according to the sequence of gene isolation [11]. Chemokines exert their effects by signaling through corresponding G protein-coupled heptahelical receptors, which are named following the aforementioned nomenclature proceeded by an “R” denoting receptor [12]. Their functions are broadly categorized into inflammation and homeostasis [11], encompassing processes such as chemotaxis, lymphocyte localization, lymphoid tissue development [13], angiogenesis [14], hematopoiesis, and degranulation [15]. Due to their chemotactic and stimulatory capacities, chemokines also play a significant role in modulating the TME (Figure 1).

### 2.1. CC Chemokines

The CC motif chemokines constitute the largest subgroup, including CCL1-28, with CCL6, 9, 10, and 12 found only in mice with no human orthologue [9,11]. Several CC chemokines, such as CCL1, CCL2, CCL5, CCL18, and CCL21, have been implicated in cancer, exhibiting both pro- and anti-tumorigenic roles [16,17,18,19].

CCL1 is a potent recruiter of regulatory T cells (Tregs) in cancer and is secreted by monocytes, macrophages, and T lymphocytes [20,21]. The expression of CCR8, the receptor of CCL1, has also been found on tumor-associated macrophages (TAMs) as well as other myeloid cell subsets [22]. Many cancers secrete CCR8 as a mechanism of immune evasion, increasing the infiltration of immune-suppressive cells [7]. The CCL1/CCR8 axis has been implicated in the progression of multiple cancers, including breast cancer and colon cancer [20,23]. The blockade of CCR8 has also been found to reduce tumor burden, making it a promising target for drug treatments [23].

The monocyte chemoattractant protein 1 (MCP-1), also known as CCL2, is an inflammatory chemokine expressed by fibroblasts, endothelial cells, and cancer cells [24]. CCL2 expression is regulated by canonical nuclear factor kappa-light-chain-enhancer of activated B cells (NF-κB) signaling in response to tumor necrosis factor (TNF) α and mTORC1-FOXK1 signaling in response to insulin [25]. The CCL2/CCR2 axis has been linked to poor prognosis in a wide range of cancers, including brain, gastric, pancreatic, breast, ovarian, prostate, esophageal, tongue, liver, colorectal, and gallbladder cancers [26,27,28,29,30,31,32,33,34]. In addition to CCL2, three other MCP chemokines exist—CCL7 (MCP-3), CCL8 (MCP-2), and CCL13 (MCP-4)—and are implicated in cancer progression [35].

CCL5, also known as RANTES (Regulated on Activation, Normal T Cell Expressed and Secreted), is a chemokine that plays a crucial role in recruiting immune cells to sites of inflammation [36]. It is produced following the activation of NF-κB, alongside CC chemokines CCL3 (MIP-1α) and CCL4 (MIP-1β) [36,37]. CCL5 recruits and binds cells expressing CCR1, CCR3, CCR4, and CCR5 such as T lymphocytes, monocytes, natural killer (NK) cells, and eosinophils [37]. The production of CCL5 is delayed, typically appearing 3–5 days after T cell activation, and is crucial in sustaining inflammatory signaling [37]. Although CCL5 can drive inflammatory responses, it is frequently implicated in promoting malignancy and is associated with poor prognosis in cancers such as prostate, breast, lung, and melanoma [38]. Due to its ability to bind multiple receptors, CCL5 has been linked to multiple pathways that mediate tumor progression, including the JAK2/STAT3, NF-κB, TGFβ, PI3K/Akt, HIF-α, and Ras-ERK-MEK pathways [39].

CCL18, also known as macrophage inflammatory protein 4 (MIP-4), pulmonary and activation-regulated chemokine (PARC), alternative macrophage activation-associated CC chemokine-1 (AMAC-1), and dendritic cell-derived CC chemokine 1 (DCCK1), is a chemokine unique to humans, with no mouse orthologue [40,41]. CCL18 is expressed in the lungs, where it contributes to asthma pathogenesis by attracting basophils that release histamine [42]. Additionally, CCL18 is produced by dendritic cells (DCs) in the germinal centers of secondary lymphoid organs, such as the tonsils, where it recruits naive T cells and mantle zone B cells, initiating the adaptive immune response [43,44]. CCL18 also serves as a maturation factor for M2-polarized macrophages, following recruitment to the tissues [40]. The upregulation of M2-like markers, such as IL-10, by CCL18 coincides with the downregulation of CCL2 expression, which may explain the M2 polarization of macrophages in the bladder following their recruitment by CCL2, although the receptor responsible for this phenomenon remains unknown [40]. CCL18 has been reported to interact with CCR6, contributing to pulmonary fibrosis [45], and with CCR3, facilitating eosinophil, CD4^+^ T helper (Th)2 cell, and basophil migration during inflammation [46]. The interactions of CCL18 with the receptor phosphatidylinositol membrane-associated transfer protein 3 (PITPNM3) have been extensively studied in breast cancer. Within the TME, TAMs secrete CCL18 which interacts with the PITPNM3 receptor of CCL18-activated normal breast-resident fibroblasts (NBFs) [47]. This interaction activates NF-κB, inducing a CD10^+^ GRP77^+^ phenotype in NBFs, which provides a survival niche for cancer stem cells through acquired chemoresistance [18,47]. Additionally, the CCL18/PITPNM3 axis has been shown to activate the JAK2/STAT3 signaling pathway, a conserved pathway involved in tumor growth and metastasis [48].

The chemokines CCL21 and CCL19 are key lymphocyte-homing ligands expressed by stromal cells in secondary lymphoid organs and on high endothelial venules [49]. The CCL19/CCL21/CCR7 axis recruits immune cells into secondary lymphoid organs, where antigen presentation and adaptive immune cell activation occur [50,51]. Both CCL19 and CCL21 can recruit CCR7^+^ cells, including naive, regulatory, and memory T lymphocytes; naive B lymphocytes; dendritic cells; and NK cells. However, CCL19 signaling is typically short-lived, as the internalization and desensitization of CCR7 following CCL19 stimulation inhibit further activation by either CCL19 or CCL21 [49]. The expression of CCR7 on non-immune cells has been linked to lymph node metastasis in various malignancies, including colon cancer, breast cancer, head and neck squamous cell carcinoma, and melanoma [19,52,53,54]. 

### 2.2. CXC Chemokines

The second largest chemokine subgroup is the CXC chemokine subgroup, which is composed of 17 chemokines [11]. CXC chemokine signaling contributes substantially to tumor progression by activating multiple tumorigenic pathways, including TGFβ, STAT3, MAPK, PI3K, and β-arrestin, which contribute to proliferation, invasion, angiogenesis, and survival [55]

The CXCR2 axis plays a fundamental role in malignancy, in part due to its ability to bind to various ligands including CXCL1, CXCL2, CXCL3, CXCL5, CXCL6, CXCL7, and CXCL8 [56]. CXCL1, also known as growth-regulated oncogene alpha (GRO-alpha) and melanoma growth-stimulatory activity (MGSA), is an inflammatory chemokine that signals through CXCR2 [57,58]. The production of interleukin (IL)-1β and TNFα during an inflammatory response increases the expression of CXCL1, which then recruits neutrophils, basophils, and CD14^+^ monocytes to the site of inflammation [58]. CXCL1 is not only involved in chemotaxis but also plays a critical role in cell activation, proliferation, senescence, and apoptosis, which collectively contribute to its oncogenic potential and complicate its role in cancer pathology [58]. Under normal physiological conditions, CXCR2 stimulation initiates a positive feedback loop in which damaged or stressed cells secrete CXCL1 through NF-κB and C/EBPβ signaling pathways [59]. This leads to further upregulation of CXCR2 via autocrine signaling and ultimately results in p53-mediated cell cycle arrest [59]. In cells with mutations in the TP53 gene, a common alteration in neoplastic cells, the positive feedback loop becomes oncogenic, resulting in increased cell proliferation instead of senescence [58]. The production of CXCL1 by various cells in response to stress and damage can exacerbate conditions within the TME, which is often hypoxic and elevated in reactive oxygen species (ROS). These factors activate the CXCL1/CXCR2 feedback loop, leading to increased recruitment of tumor-associated neutrophils (TANs) and granulocytic myeloid-derived suppressor cells (G-MDSCs) [60,61]. TAMs are also capable of producing CXCL1, which was found to promote metastasis in breast cancer via NF-κB/SOX4 signaling [62]. Cancer cell production of CXCL1 was found to induce CD8^+^ T cell exhaustion through the recruitment of polymorphonuclear (PMN)-MDSCs in gastric cancer [63]. Stimulation of CXCR2 has also been found to induce tumor growth migration, invasion, metastasis, epithelial-to-mesenchymal transition (EMT), and chemoresistance in multiple malignancies [64,65,66]. 

CXCL8 was first discovered as the monocyte-derived neutrophil chemotactic factor (MDNCF) after purification from LPS-activated peripheral blood mononuclear cell (PBMC) supernatant, then renamed neutrophil-activating peptide (NAP-1) due to its effect on neutrophils [10]. Based on its capacity to also recruit a subset of T lymphocytes, NAP-1 was renamed interleukin IL-8, which was then changed to CXCL8 following the discovery of MCP-1 [10]. As one of the first discovered chemokines, the role of CXCL8 in the inflammatory response has been extensively studied. The release of CXCL8 by macrophages and tissues in response to infection triggers the recruitment and activation of neutrophils via CXCR1 and CXCR2 [67]. Following the inflammatory response, CXCL8 also participates in tissue repair by promoting angiogenesis, proliferation, and migration of CXCR2-expressing endothelial cells [67]. Upregulation of CXCL8 expression on cancer cells is common across malignancies and is correlated with tumor progression and an overall worse prognosis [68,69]. CXCL8 expression on cancer cells can be induced by inflammatory cytokines such as TNFα and IL1β, as well as by stress conditions including hypoxia, oxidative stress, and exposure to chemical agents [69]. This induction activates tumorigenesis-related pathways through the binding of CXCL8 to its receptors, CXCR1 and CXCR2 [68,69]. 

Chemokines CXCL9 (also known as monokine induced by gamma interferon, or Mig) and CXCL10 (also known as interferon-γ-induced protein 10, or IP-10) are secreted by various cells, including monocytes and fibroblasts, in response to stimulation with interferon (IFN)-γ [70,71]. CXCL9 and CXCL10 share 37% amino acid identity, and their genes are found adjacent on chromosome 4q21.21, suggesting a close evolutionary relationship [72,73]. Both play crucial roles in chemotaxis by binding to the receptor CXCR3 of IL-12-activated T cells, NK cells, DCs, and macrophages [72,74,75,76]. The CXCL9/CXCL10/CXCR3 axis plays a role in recruiting tumor-infiltrating lymphocytes (TILs) to tumors across various cancers, indicating a potential anti-tumor function [77]. In studies involving Burkitt’s lymphoma in nude mice, CXCL9 and CXCL10 have been reported to induce tumor necrosis. Additionally, the expression of these chemokines within Burkitt’s lymphoma cells has been shown to decrease their capacity to form subcutaneous tumors [78].

CXCL12, also known as stromal cell-derived factor-1 (SDF-1), is a homeostatic chemokine that binds CXCR4 and the orphan receptor, ACKR3 [79]. Initially termed the pre-B cell growth factor, CXCL12 is widely expressed in stromal cells, secondary lymphoid organs, epithelial cells, circulating stem cells, and hematopoietic progenitors [79]. The fundamental role of the CXCL12/CXCR4 axis in embryonic development has been demonstrated in CXCR4 knockout and CXCL12 knockout mice, with CXCR4 deletion found to be the only lethal chemokine deletion [80,81]. Deletion of CXCL12 in mice embryos demonstrated nearly complete ablation of the pro- and pre-B cell population in the fetal liver and an absence of myelopoiesis in the bone marrow during late embryonic development, as well as cardiogenic and neurogenic defects [80]. The highly conserved genomic sequence of CXCL12 and CXCR4, as well as the binding specificity of CXCR4 to CXCL12, further underscores their importance in development and survival [79]. Postnatal functions of CXCL12 are also critical in homeostasis and tissue repair. As a key chemoattractant, CXCL12 mediates hemopoietic stem cell migration to the bone marrow or injured tissue [82]. In malignancies, the chemotactic and homeostatic functions of CXCL12 support tumor progression and invasion [82]. Various cancers, including glioblastoma, prostate cancer, breast cancer, lung cancer, multiple myeloma, cervical cancer, and pancreatic cancer, exhibited increased proliferation in response to CXCL12/CXCR4 in a concentration-dependent matter [83,84].

B cell-attracting chemokine, or CXCL13, is highly expressed on follicular DCs within B cell follicles and germinal centers of secondary lymphoid organs [85,86]. CXCL13 induces an increase in calcium influx and the chemotaxis of B cells via BLR-1, also known as CXCR5 [87]. The CXCL13/CXCR5 axis plays a crucial role in B cell maturation by recruiting B cells, CD4^+^ follicular T helper cells (Tfhs), follicular Tregs, follicular CD8^+^ T cells, and natural killer T (NKT) follicular helper cells to the germinal center. This recruitment is essential for modulating affinity maturation and isotype switching [88]. CXCL13 expression has been reported to hold pro-tumorigenic roles in cancers such as central nervous system lymphoma, follicular lymphomas, and colon carcinoma [89,90,91]. CXCL13 induces lymphotoxin production by naïve B cells and directly recruits lymphoid tissue-inducer (LTi) cells. Both functions contribute to the formation of tertiary lymphoid structures (TLSs), ectopic aggregates resembling secondary lymphoid organs often found in areas of inflammation [92,93]. In ovarian cancer, CXCL13 was correlated with the formation of TLSs and improved survival [93]. Given that CXCL13 exhibits both pro- and anti-tumor properties, further research is needed to elucidate its role in various cancers and to define its potential therapeutic applications.

Phosphatidylserine and oxidized lipoprotein (SR-PSOX or CXCL16) is a widely expressed chemokine found in multiple organs, including the pancreas, heart, liver, testis, thymus, spleen, kidney, lung, and prostate, as well as on immune cells [94,95]. Alternative splicing and post-transcriptional modifications result in the identification of multiple isoforms of CXCL16 [95]. CXCL16 cleavage generates soluble CXCL16, which modulates the recruitment of inflammatory immune cells through the CXCR6 receptor [96,97]. The role of CXCL16 varies across malignancies; for instance, CXCL16 and its receptor CXCR6 are upregulated in breast cancer but downregulated in renal cell carcinoma [94,98,99]. Elevated expression of CXCL16 in tumors or serum has prognostic significance in cancers such as colon cancer [94]. 

### 2.3. CX_3_C Chemokines

CX_3_CL1, also known as fractalkine, is the only member of the CX_3_C subgroup [100]. As the most abundantly expressed chemokine in neurons, CX_3_CL1 plays a fundamental role in the formation of neuronal networks, cognitive development, and recruitment of microglia [100]. CX_3_CR1 expression is not limited to microglia; it is also found in various immune cell subsets, including NK cells, T cells, dendritic cells, and monocytes [101]. Through alternative splicing, CX_3_CL1 can exist as either a soluble protein or a membrane-bound protein, expressed on the luminal surface of the endothelium in response to stimulation by inflammatory cytokines such as TNFα, IFNγ, and IL-1 [102,103]. In addition to its chemotactic functions, the CX_3_CL1/CX_3_CR1 axis is a mediator of inflammation and survival [104,105,106]. In ankylosing spondylitis, CX_3_CL1 was found to induce M1 polarization of macrophages through the NF-κB pathway, and in a murine model of systemic candidiasis, CX_3_CR1 inhibited the caspase-dependent apoptosis of renal macrophages through Akt activation [106].

The chemotactic capacity of CX_3_CL1 contributes to its anti-tumorigenic functions; however, evidence also suggests that CX_3_CL1 may have pro-tumorigenic roles. In colorectal cancer, elevated tumor expression of CX_3_CL1 has been correlated with improved prognosis and increased infiltration of CX_3_CR1-expressing cytotoxic NK and T cells [107]. In contrast, CX_3_CL1/CX_3_CR1 signaling has been associated with neurotropic metastasis in pancreatic ductal adenocarcinoma, increased cancer cell proliferation in ovarian cancer, and enhanced invasion and migration in lung cancer through activation of Src/FAK pathways [108,109,110]. Additionally, CX_3_CL1/CX_3_CR1 signaling induces paclitaxel resistance in gastric cancer through RhoA signaling and promotes migration and resistance to anti-PD-1 therapy in colon cancer, and the blockade of CX_3_CL1/CX_3_CR1 has proven effective in both cases [111,112].

### 2.4. XCL Chemokines

XCL1, also known as lymphotactin, and XCL2 are members of the XC motif chemokine family and signal through the receptor XCR1 [113]. Both XCL1 and XCL2 are pro-inflammatory cytokines that are constitutively expressed by NK cells, though XCL1 has also been detected on Th1-polarized CD4^+^ T cells and CD8^+^ T cells [113,114]. The functional roles of XCL1 and XCL2 remain unclear; however, increased production of XCL1/2 by NK cells in head and neck squamous cell carcinoma, melanoma, and triple-negative breast cancer has been associated with improved patient survival, likely through the recruitment of conventional type 1 dendritic cells (cDC1) [115]. However, other studies have shown a correlation between elevated XCL1 and PD-L1 expression in squamous cell carcinoma, contributing to the exhaustion of intratumoral T cells [114,116,117]. 

## 3. Chemokines in Bladder Tumor Development and Progression

### 3.1. Chemokines Mediate the Recruitment of Immunosuppressive Immune Cell Subsets

Chemokines play a crucial role in bladder tumor development and progression by orchestrating the recruitment and interaction of various immune and stromal cells within the tumor microenvironment (Figure 2, Table 1). While the precise mechanisms remain under investigation, it is well established that the recruitment of TAMs and the interplay between TAMs, cancer-associated fibroblasts (CAFs), and cancer cells are features across different malignancies and contribute significantly to tumor metastasis and progression [24]. Specifically, bladder tumors leverage the upregulation of chemokines such as CCL2 and CXCL8 to attract immunosuppressive immune cells, thereby enhancing tumor growth and facilitating a more aggressive disease course [24]. Additionally, recruited immunosuppressive MDSCs expressing PD-L1 can further contribute to the exhausted phenotype of T and NK cells in the bladder TME through PD-1/PD-L1 binding [118]. 

The expression of CCL2 in BCa is found to play a significant role in modulating the TME through the recruitment of pro-tumor immune cells including TAMs and MDSCs [16]. CCL2 may also play a role in polarizing recruited macrophages toward an M2 phenotype [136]. Macrophages polarize into either a pro-inflammatory phenotype with high CCR2 expression in response to granulocyte–macrophage colony-stimulating factor (GM-CSF) stimulation or into an anti-inflammatory M2 phenotype characterized by elevated levels of CCL2 and CCL8 in response to macrophage colony-stimulating factor (M-CSF) [137,138]. Bladder tumors have been found to express high levels of M-CSF, capable of recruiting macrophages that express CCL2 and inducing M2 polarization [139]. 

CXCL8 has also been correlated with increased infiltration of TAMs, neutrophils, and an immunosuppressive TME [130,140]. In vitro analysis of BCa cell lines and normal urothelial cell lines showed varying levels of CXCL8, with the T24 BCa cell line expressing the highest level of CXCL8 and IL6 [140]. Additionally, co-incubation of BCa cells with PBMCs from healthy donors resulted in elevated levels of Tregs, suggesting a CXCL8-mediated induction of Treg differentiation [140]. The increased Treg infiltration of the tumor was independent of TGFβ signaling, suggesting a systemic distribution of CXCL8 and IL6 produced by cancer cells, which results in increased neutrophil production, Foxp3 upregulation on T lymphocytes, and tumor trafficking through CXCR1 [140]. 

Neutrophils, specifically, are highly receptive to CXCL8 recruitment and activation due to their high expression of CXCR1 and CXCR2 [129]. In BCa, elevated levels of TANs and neutrophil extracellular traps (NETs) have been associated with invasion, metastasis, and tumor progression and growth [127]. NETosis, a unique form of cell death specific to neutrophils, involves the degradation of the neutrophil cytoskeleton by proteases, leading to degranulation and the release of dsDNA [127]. This process forms adhesive, net-like structures that immobilize microbes [141]. The formation of NETs in BCa is significantly correlated with CXCL8 expression and worse prognosis [142]. In cancer, NETosis has been linked to tumor cell proliferation through the activation of NFκB and it also directly modulates the mitochondrial metabolic activity of cancer cells [143]. Murine models investigating the formation of NETs in bladder cancer have demonstrated that these structures create a barrier between the tumor and the stroma, inhibiting CD8^+^ T cell infiltration and directly contributing to the tumor’s resistance to radiation [144]. CXCL1 and CXCL2 also recruit TANs via CXCR1 and CXCR2 [125].

CCL5 signaling also contributes to the recruitment of TAMs [120]. Co-culturing THP-1 monocytic cells, which resemble an M2-like macrophage phenotype, with the TCCSUP BCa cell line demonstrates the crosstalk between BCa cells and TAMs. Exposure to BCa cells results in increased CCL5 production by TAMs, which subsequently enhances the infiltration, invasion, and proliferation of the BCa cells [120]. Additionally, co-culture increased the expression of CXCL8 on BCa cells, which is known to recruit TAMs and TANs [120,130,140]. These data suggest a possible feedback loop of chemokine production that further enhances TAM recruitment, tumor progression, and continued chemokine production [120].

### 3.2. Chemokines Also Recruit Activated Cytotoxic Immune Cells

Chemokines have also been implicated in the increased recruitment of cytotoxic immune cells, increasing anti-tumor immunity. A multiplex staining analysis of patient-derived samples revealed that BCa tissue contained higher levels of XCL2^+^ cells compared to adjacent para-cancerous tissue. These XCL2^+^ cells are likely indicative of immune infiltrates [145]. This observation is supported by the increased mRNA expression of XCL2 on NK cells co-incubated with BCa cells, as well as elevated levels of XCL1 [133]. While the roles of XCL1 and XCL2 in the BCa TME are still being explored, studies have found that XCL1 and XCL2 increase the infiltration of XCR1^+^ dendritic cells and M1-polarized macrophages [115]. 

Co-culturing NK cells with BCa cells was also found to increase the expression of CCL1, CCL2, CCL20, and CXCL16 [133]. In addition to the direct NK-mediated cytotoxicity, the elevated expression of CCL1, CCL2, and CCL20 on NK cells increased T-cell chemotaxis in vitro [133]. While it is challenging to definitively attribute an anti-tumor role to CXCL16-expressing NK cells, they may contribute to the activation of CXCR6^+^ immune cells, such as invariant NKT cells and intratumoral CD8^+^ T cells, potentially enhancing their anti-tumor capacity [146,147].

The formation of TLSs is associated with improved patient prognosis and treatment response in BCa, likely due to their role in recruiting and activating immune cells [148]. The exact mechanism of TLS formation remains unclear. Though many chemokines have been implicated, the presence of TLSs in BCa is correlated with an elevated expression of CXCL13, a key modulator of TLS formation [132,148]. In an inflammatory setting, stromal cell activation induces the recruitment of LTis, or other lymphotoxin (LTα1β2)-expressing immune cells such as innate lymphoid cell (ILC)3, Th17 CD4^+^ T cells, B cells, and macrophages as a result of IL-13, IL17, and IL-22 produced by leukocytes [149]. Stromal cells act as lymphoid tissue organizers by interacting with TNF- and LTα1β1-expressing immune cells via TNF receptor 1 (TNFR1) and lymphotoxin β receptor (LTβR). They secrete immune-recruiting chemokines such as CCL12, CCL19, CCL21, and CXCL13, as well as VEGF-C and IL-7, which regulate the formation of high endothelial venules [149]. Mature TLSs contain functional germinal centers and various immune cells, including T follicular helper (Tfh) and follicular dendritic cells, which regulate antigen presentation, affinity maturation, and isotype switching of B cells [149]. The role of CXCL13 in both the induction of TLS formation and immune cell recruitment makes it a key marker of TLSs and a prognostic marker for immunotherapy response in BCa [132]. 

### 3.3. The Function of Chemokines in Bladder Cancer Is Sex-Dependent

An extremely understudied axis in BCa diagnosis and treatment is the relation between sex and immune response. The dichotomy of the male versus female immune system also remains a mystery, though epigenetic studies have found a possible cause for the difference in immune response: the X chromosome. Genetically, females inherit an X chromosome from each parent, while males carry a maternal X chromosome and a paternal Y chromosome [150]. To accommodate for the extra X chromosome and maintain gene dosage, epigenetic modifications inactivate an X chromosome at random, creating a Barr body [150]. Studies have found that approximately 15% of genes escape the inactivation, including immunity-related genes such as toll-like receptor 7 (TLR7), Burton tyrosine kinase, CXCR3, IL-1 receptor-associated kinase 1, and CD40 ligand [150,151]. Additionally, in mice, the female humoral response results in higher IgG production because of elevated TLR7 expression on B cells [151]. These factors may also contribute to the sex-associated immune responses in cancer.

It has been suggested that CCL5 expression and resulting prognosis are sex-dependent. As BCa is four times more likely to affect males vs. females, so differential expression of CCL5 and the resulting outcomes are important factors to consider when providing treatment to male vs. female patients [152]. In male patients with BCa, CCL5 expression was found to be decreased and associated with a worse prognosis, while in female BCa patients, CCL5 levels were unchanged [153,154]. Interestingly, an orthotopic BCa model using female mice showed a significant and sustained increase in the expression of CCL5 from the onset of malignancy [155]. It remains challenging to conclusively delineate the role of CCL5 in BCa between males and females, partly due to the significantly smaller sample size of females with BCa and the lack of orthotopic BCa models using male mice. These differences may be attributed to variations in immune responses between sexes. Previous studies in females have shown that estradiol treatment increases IFNγ expression and is associated with higher activation capacity of immune cells and a more pronounced inflammatory response [152]. These findings may explain the increased levels of CCL5 in females specifically. BCG, an intravesical treatment that stimulates a localized inflammatory response, significantly increases the production of CCL5, further suggesting that the role of CCL5 in BCa is influenced by the immune environment [156]. 

Although CXCL13 has been identified as a promising biomarker for indicating favorable clinical outcomes, high CXCL13 expression is correlated with high-grade tumors in female patients with NMIBC [157]. In female BCa, the tumor microenvironment shows significant increases in B cell recruitment via CXCL13 expression, along with higher PD-L1 expression, increased infiltration of CD163^+^ M2-like TAMs, and differences in immunoregulatory gene expression. These factors highlight distinct differences in BCa between male and female patients [157]. These differences contributed to resistance to immunotherapeutic therapies and had shorter progression-free and recurrence-free survival, which indicate the importance of gender when administering immunotherapy [157].

### 3.4. Chemokine Lymph Node Homing Induces Lymph Node Metastasis in Bladder Cancer

In BCa, the CCR7/CCL21 axis increases proliferation, invasion, and lymph node metastasis and decreases the apoptosis of cancer cells [123,124]. Immunohistochemistry (IHC) staining of human bladder tumor samples revealed elevated expression of CCR7 in BCa, with 64.52% of the cohort exhibiting high levels of CCR7 [124]. CCR7 expression was also found to correlate with a worse overall prognosis in patients, showing increased levels in those with lymph node metastasis, as well as higher tumor stage and grade [124]. Analysis of microvessel densities (MVDs) and microlymphatic vessel densities (MLVDs) revealed that high CCR7 expression significantly correlates with both MVDs and MLVDs [124]. 

The CXCL1/CXCL8/CXCR2 axis is implicated in increased lymph node metastasis in BCa. IHC staining of patient tumor tissue demonstrated an abundance of CD66b^+^ TANs in higher-grade BCa and MIBC [126]. Additionally, increased BCa metastasis to the popliteal lymph node was observed in a murine model, which was mitigated by TAN depletion [126]. Mechanistic analysis of isolated neutrophils exposed to culture media from T24 and UMUC3 BCa cells revealed a significant increase in production of VEGF-A and matrix metalloproteinase (MMP)-9 through CXCL1/CXCL8/CXCR2-mediated activation of the ERK/JNK signaling pathway, which induces lymphangiogenesis and lymph node metastasis [126]. Lymph node metastasis is the primary mechanism through which BCa spreads, leading to poor prognosis and an increased mortality rate within 5 years compared to non-metastatic cases [158]. Understanding the chemotactic roles of chemokines is crucial for developing targeted therapies.

### 3.5. Expression of Chemokine Receptors on Bladder Cancer Cells Activates Pro-Tumor Signaling Pathways

The role of chemokines in BCa is not limited to recruiting and modulating the phenotypic expression of immune infiltrates in BCa. Upregulation and de novo expression of chemokine receptors on BCa cells, including CCR5, CXCR2, and CCR7, is often seen during tumorigenesis and directly modulates bladder tumor growth, survival, and EMT (Figure 3) [121,123,124,159].

CCL5 binds with a high affinity to the G-protein-coupled receptor, CCR5, which is upstream of multiple pro-tumor signaling pathways including PI3K/Akt and JAK2/STAT3 [36,121]. CCL5/CCR5 mediates the activation of JAK2/STAT3 in the BCa cell lines, T24, and J82 [121]. The knockdown of CCL5 in the T24 cell line inhibited proliferation, invasion, and migration in vitro, while CCL5 overexpression increased these processes [121]. Furthermore, the expression of CCL5 was consistent with the phosphorylation of both JAK2 and STAT3 and illustrated the role of JAK2/STAT3 in proliferation, invasion, and migration by chemically inhibiting JAK2 phosphorylation in vitro [121]. CCL5 plays a role in BCa progression, with higher-stage bladder tumors showing increased infiltrating TAMs. Exposure to BCa cells increases CCL5 expression on TAMs, which enhances the infiltration, invasion, and proliferation of BCa cells [120]. 

Conversely, CCR5 expression on BCa cells promotes invasion and migration. While the exact mechanism is not fully understood, pathways involved in CCR5/CCL5 signaling identified in other malignancies may also be relevant to BCa [121]. For example, in melanoma, CCR5 was found to induce EMT through the secretion of TGFβ1 by CCR5^+^ cancer cells, triggering non-canonical TGFβ-mediated activation of the PI3K/Akt/GSK3β signaling pathway [160]. The increased invasive capacity of CCR5-expressing cancer cells was also described in basal breast cancer, colorectal cancer, pancreatic cancer, and prostate cancer [38,161,162,163]. CCR5 expression on osteosarcoma has also been found to induce angiogenesis through the production of VEGF via PKCδ/c-Src signaling and HIF-α accumulation [17]. 

CCR7 is a G-protein-coupled receptor (GPCR) that, beyond its role in lymph node homing, has additional roles in BCa [49]. In vitro studies show that CCR7/CCL21 binding activates specific pathways in BCa cells [123,124]. Treatment of both UM-UC3 and T24 BCa cells with CCL21 was found to significantly increase in vitro migration and invasion, and the knockdown of CCR7 expression abolished the CCL21 effect [123,124]. Activation of CCR7 with CCL21 increased BCa cell proliferation and the expression of MMP-2, MMP-9, VEGF-C, and Bcl-2, while decreasing Bax expression, suggesting a role for CCR7/CCL21 in BCa cell survival and EMT [123]. Treatment of BCa cells with CCL21 was also found to activate the MEK/ERK1/2 pathway, which was abrogated by CCR7 knockdown [124]. 

CCL18/CCR8 signaling induces EMT through the upregulation of MMP-2 and VEGF-C and by decreasing E-cadherin expression [122]. Migration and invasion of BCa cells are dependent on CCL18/CCR8 interaction: inhibition with R423 or CCR8 shRNA transfection impaired these processes compared to control cells [122]. Furthermore, CCL18 has been identified as a biomarker for detecting BCa. ELISA analysis of urine samples from healthy individuals and BCa patients revealed significantly higher urinary concentrations of CCL18 in BCa patients compared to healthy controls [164]. 

In BCa, CXCL8 is consistently overexpressed and has been correlated with recurrence, progression, higher tumor grade, and poor prognosis [165,166,167,168,169]. Although studies on CXCR1 and CXCR2 expression in BCa cells and their prognostic implications are lacking, both receptors are expressed by normal urothelial cells [170]. Additionally, autocrine signaling of CXCL8 has been confirmed in both healthy and malignant urothelial cells [170,171]. In healthy urothelial cells, CXCL8-dependent cell survival was found to signal specifically through CXCR1 via Akt signaling in vitro, and the inhibition of CXCL8 expression in urothelial cells resulted in cell surface expression of CXCR1 with no changes in *CXCR1* mRNA levels [170]. Furthermore, induced malignancy in healthy urothelial cells in vitro demonstrated cell surface expression of CXCR1 [171]. These findings indicate that CXCR1 is primarily localized intracellularly in healthy urothelial cells and may be upregulated and expressed under conditions that compromise cell survival, such as the lack of CXCL8-dependent cell survival signaling or in malignancy, where CXCR1 overexpression could drive tumorigenesis. Additionally, CXCR1 internalization upon CXCL8 binding explains the low baseline cell surface expression of CXCR1 in urothelial cells [171]. However, this phenomenon remains inadequately understood and requires further investigation.

Other studies have also implicated CXCL8 in tumor progression. IHC staining of clinical samples revealed elevated CXCL8 levels in higher-stage and -grade BCa tumors. This elevation is correlated with increased VEGF, MMP-9, and TAMs, as well as decreased E-cadherin levels [130]. CXCL8 produced by TAMs was found to increase BCa invasion, angiogenesis, and migration in vitro, as well as increase levels of MMP-9 [130]. The upregulation of the tight junction protein, occludin, was also found to upregulate CXCL8 expression through STAT4 activation, which induced angiogenesis through CXCL8/STAT3 activation [128]. 

CXCR2 expression on BCa cells can also contribute to migration and survival. As we previously discussed, the expression of CXCR2 on cancer cells in response to stress modulates tumorigenesis through the inhibition of apoptosis [59]. Specifically, CXCL5/CXCR2 signaling was found to increase the migratory and invasion capacities of BCa cells through MMP-2 and MMP-9 [159]. Both CXCL5 and CXCR2 are expressed on BCa cells, with CXCL5 expression being notably higher in tumors of increased grade [159]. The upregulation of MMP-2 and MMP-9 through CXCL5/CXCR2 signaling is mediated by the activation of the PI3K/Akt pathway and is specific to CXCL5 [159]. 

The recruitment of TAMs by CXCL12 is being investigated as a mechanism driving tumor proliferation and invasion. In vitro, the knockdown of CXCL12 significantly reduced TAM chemotaxis and inhibited the overproduction of cytokines, including IL-10, VEGF, IL-4, and TGFβ [131]. The transcription factor SPI-1 is linked to CXCL12 expression, as the inhibitory effects of CXCL12 knockdown were reversed by adding SPI-1 in vitro [131]. In a xenograft murine model, SPI1/CXCL12 was associated with increased tumor weights when SPI1 was present, while CXCL12 knockdown led to decreased tumor weights [131].

CXCL16 exhibits both pro-tumorigenic and anti-tumorigenic properties in BCa. Transcriptome analysis of high-grade urothelial carcinomas revealed that CXCL16 expression is significantly associated with PD-L1-mediated immune evasion, indicating its role in promoting tumor immune tolerance and progression [134,172]. On the other hand, CXCL16 expression is upregulated in NK cells cultured in vitro with BCa cells, suggesting an anti-tumorigenic role in T cell chemotaxis [133]. CXCL16 gene upregulation was statistically associated with longer disease-free survival (DFS) in BCa [172]. Conversely, CXCL16 exerted pro-tumorigenic effects in BCa by activating the ERK1/ERK2 signaling pathway and promoting tumor proliferation in response to IFNγ [134].

## 4. Prognostic Relevance of Chemokines in Bladder Cancer

Studies have found that CCL2 expression correlates with higher tumor stage, with muscle-invasive BCa showing greater CCL2 expression than non-muscle-invasive BCa [173,174]. Specifically, CCL2 expression was correlated with shorter OS and increased mortality in T2 and luminal-type bladder tumors [119]. Although the mechanistic signaling pathways regulating CCL2 expression and function in BCa remain largely unknown, CCL2 is known to bind to CCR2 and signal through the PI3K/Akt, MAPK/p38, and JAK/STAT3 pathways, which are involved in tumorigenesis [16]. CCL2 expression is upregulated by the heat shock protein (HSP) 47, which is highly expressed in BCa [175]. HSP47 knockdown in MGH-U3 and T24 BCa cell lines significantly decreased angiogenesis and HUVEC migration in vitro, and reduced bladder tumor burden in vivo [175]. Blocking CCR2 significantly decreased the phosphorylation of ERK1/2, a downstream mediator of MAPK signaling [175]. Overall, these findings suggest that malignant BCa cells with increased HSP47 expression increase the expression of various pro-angiogenic factors, including CCL2, which then increases angiogenesis, migration, and tumor burden [175]. The CCL2/CCR2 axis has also been found to induce BCa cell migration independent of ERK signaling. In both in vitro and in vivo models, CCL2/CCR2 signaling was found to increase paxillin phosphorylation through PKC activation [174]. 

CCL5 has also been connected to tumor proliferation, invasion, migration, and reconstruction of the extracellular matrix in BCa; however, prognostic studies have found CCL5 to be anti-tumorigenic as well [121]. A clinical study on MIBC revealed distinct prognostic outcomes based on the expression of CCL5 in tumor cells versus infiltrating immune cells. Tumor cells expressing CCL5 were associated with worse DFS, overall survival (OS), and relapse-free survival (RFS), especially in more advanced stages. Additionally, CCL5 expression in these tumor cells was linked to an increased risk of both overall and disease-specific death [176]. In contrast, CCL5 expression on infiltrating immune cells was associated with longer OS, DFS, and RFS. This suggests that CCL5 may play a crucial anti-tumor role by facilitating the recruitment of immune cells, despite its paradoxical association with disease progression [176]. Current data suggest that the impact of CCL5 may be influenced by the specific cells expressing CCL5 or its receptor, CCR5. While we have discussed the prognostic differences between CCL5 expression on BCa cells and immune cells, the expression of CCR5 is also crucial. Due to the chemotactic properties of CCL5, immune cells expressing CCR5 are increasingly recruited to the BCa TME, which is correlated with survival outcomes [177,178,179]. A unique subset of CCR5^+^CD66b^+^ tumor-infiltrating neutrophils accumulates in bladder tumors in MIBC. These neutrophils show increased production of IFNγ and improve the tumor response to the anti-PD-1 immunotherapy pembrolizumab [179].

Bioinformatic analyses using multiple databases have shown that mRNA expression of several chemokines, including CCL4, CCL5, CCL14, CCL19, CCL21, and CCL23, is downregulated in BCa [153]. Interestingly, CCL2 expression was decreased in BCa tissue, and this was correlated with poor DFS, conflicting with other studies [153,174,175]. In a similar study focusing on CXC chemokines, mRNA levels of CXCL1, CXCL5, CXCL6, CXCL7, CXCL9, CXCL10, CXCL11, CXCL13, CXCL16, and CXCL17 were upregulated in BCa, while CXCL2, CXCL3, and CXCL12 were downregulated [172]. Of the upregulated genes, CXCL1, CXCL6, CXCL10, CXCL11, and CXCL13 upregulation significantly correlated with poor OS, along with upregulated CXCL12 [172].

Consistent with the bioinformatic analysis, other studies have found that CXCL1 is increased in BCa cells and the urine of BCa patients. This suggests that CXCL1 could serve as a potential biomarker for higher-grade and higher-stage tumors [180,181,182]. In tissue IHC staining, CXCL1 expression was significantly higher in high-grade tumors compared to low-grade tumors. Additionally, CXCL1 was markedly absent in benign bladder tissue, highlighting a significant difference in expression between BCa tissues and benign samples [182]. CXCL1 was also associated with reduced survival [180,182]. CXCL1 urine concentrations were also increased significantly in BCa compared to subjects without BCa; however, no differences were found between stages [181]. 

CXCL12 and CXCR4 levels are significantly elevated in BCa tissue compared to normal bladder tissue. Additionally, these levels are higher in tumors with higher invasion and poorer differentiation, regardless of age or gender [84,183]. Increased CXCL12 in BCa was correlated with reduced OS [84,131]. CXCR2 is abundantly expressed on neutrophils. IHC staining for neutrophil-specific markers MPO^+^ and CD66b^+^ in BCa tumors shows a positive correlation with higher grades, increased muscle invasion, and poorer OS [125,126]. In a gene expression study using quantitative real-time (qRT)-PCR, CXCR6 levels, the receptor for CXCL16, were significantly elevated in BCa compared to benign bladder diseases. Although CXCL16 expression did not differ significantly between BCa and controls, IHC staining showed increased immunoreactivity for both CXCL16 and CXCR6 in BCa [184]. 

In BCa, CX_3_CL1 was expressed significantly higher in BCa tissues than in normal tissues and there were significantly higher levels of CX_3_CL1 in the peripheral blood of BCa patients [185]. The role of CX_3_CL1 has been linked to a higher risk of disease recurrence and cancer-specific death. In patients with BCa, positive CX_3_CL1 expression is significantly associated with advanced tumor stage, larger tumor size, higher grade, and the presence of metastasis [135]. In a xenograft mouse model, CX_3_CL1 knockdown cells had significantly lower tumor weights compared to the negative control. Additionally, CX_3_CL1 was found to significantly promote the migration of T24 cells, further supporting the role of CX_3_CL1 in tumor progression [135].

As a mediator of inflammatory response, BCG has also been found to influence chemokine expression. Muthuswamy et al. show a significant increase in neutrophil- and MDSC-recruiting chemokines CXCL8 and CCL22 following BCG treatment in an ex vivo model, with no effect on CTL-targeting chemokines [186], while another study found increased production of CCL5 following intravesical instillation of BCG [156]. These conflicting results may be due to the specific cell populations observed in each study; however, they may also be attributed to BCG failure in certain patients, highlighting the unique responses of individuals to BCG treatment. Elevated levels of CXCL12, CXCR4, and CXCR5 following BCG treatment have been linked to BCG failure by fostering an immunosuppressive environment [187,188]. Elevated levels of CXCL12, alongside CXCL8 and CCL22, recruited Tregs, neutrophils, and MDSCs, shifting the immune infiltrates from cytotoxic to immunoregulatory subsets [187,188]. This shift from CCL5/CXCL9/CXCL10 to CXCL8/CXCL12/CCL22 production was attributed to NFκB-mediated induction of COX2/PGE2/EP4, along with upregulation of IDO1 and IL-10, further contributing to the immunosuppressive tumor microenvironment [187,188]. The standard of care in the treatment of NMIBC requires multiple instillations of intravesical BCG, which results in an expansion of exhausted cytotoxic immune cells [189]. An in vivo analysis of male and female mice with N-butyl-N-(4-hydroxybutyl)nitrosamine (BBN)-induced BCa, simulating spontaneous malignancy following carcinogen exposure, showed a female-specific accumulation of atypical B cells and TLS following BCG treatment, along with an increase in TLSs and B cell-associated chemokines CXCR4, CXCR5, and CXCL13 [189]. Notably, the depletion of B cells in female mice coupled with BCG treatment increased both Th1 and Th2 cytokines, as well as IgG antibodies following cessation of B cell depletion compared to the male counterparts [189]. Additionally, spatial immune profiling of the murine bladders demonstrated a higher infiltration of PD-L1 expressing myeloid cells in female mice treated with BCG, with a significant increase in splenic CD11b^+^ myeloid cells following only one treatment of BCG [189]. An analysis of bladder tissue derived from patients with NMIBC treated with BCG revealed elevated expression of immunoregulatory and atypical B cell markers in areas of mature TLSs, specifically in bladders from BCG non-responders, suggesting an elevation in immune cell exhaustion associated with BCG-induced inflammation [189]. Although TLSs are often associated with a better prognosis in BCa, the presence of intratumoral atypical B cells may contribute to the exhaustive phenotype of immune infiltrates and BCG failure [189]. Together, the expression of intratumoral CXCL13, CXCR4, and CXCR5 may be used as biomarkers for possible BCG failure before treatment.

Additional studies have explored the correlation of CXCL8 levels with BCa prognosis. CXCL8 expression is consistently elevated in BCa and patient urine and is associated with increased infiltration of TAMs [130,190]. A survival study of BCG-treated patients showed a four-fold increase in BCa recurrence in patients who had high urine levels of CXCL8 prior to treatment [191]. Interestingly, patients with high CXCL8 expression responded better to combination therapy with gemcitabine, cisplatin, and pembrolizumab compared to neoadjuvant pembrolizumab alone [192]. With immune-targeting treatments, such as BCG and atezolizumab, CXCL9 and CXCL10 serve as molecular markers for predicting a good therapeutic response and OS in BCa patients [155,193,194,195]. 

Low CXCL9 expression is associated with a poorer prognosis in younger patients (<71) with non-muscle-invasive bladder cancer, correlating with a reduced recurrence-free survival rate [196]. Conversely, tumor-associated dendritic cells (TADCs) produce significantly higher levels of CXCL9. In vitro, CXCL9 production by TADCs increased PD-L1 expression on BCa T24 cells, thereby inhibiting anti-tumor T cell responses and promoting tumor growth [197]. Another bioinformatics study found that CXCL10 is upregulated in BCa. Elevated levels of CXCL10 mRNA were significantly associated with poor OS, suggesting that CXCL10 plays a crucial role in the proliferation and development of BCa [172]. The correlation between CXCL10 and patient survival is inconsistent across multiple studies, which may be attributed to genomic instability caused by differential CXCL10 expression [198]. The expression of CXCL10 is elevated in tumor tissue compared to normal bladder tissue, specifically in high-grade, non-papillary tumors [198]. Elevated levels of CXCL10 in bladder tumors were correlated with a higher tumor mutational burden, including significantly increased mutations in TP53 and RB1 [198]. Additionally, tumors with high CXCL10 expression correlated with increased infiltration of anti-tumor immune subsets, including CD4^+^ and CD8^+^ T cells and M1 macrophages, as well as elevated levels of the exhaustion markers CTLA4 and PD-1 [198]. The combined tumor mutational burden and immune-rich characteristics of CXCL10-high tumors make them attractive targets for ICI therapy and suggest that CXCL10 could serve as a potential biomarker for immunotherapy success in BCa.

Several studies have proposed CXCL13 as a biomarker associated with a favorable prognosis. The expression of CXCL13 in baseline tumor tissues, along with an ARID1A mutation in tumor cells, has been identified as a predictor of successful clinical responses in metastatic urothelial carcinoma patients treated with nivolumab, an immune checkpoint therapy, through whole-exome sequencing [199]. To further validate CXCL13 as a reliable biomarker, CXCL13^−^/^−^ tumor-bearing mice were treated with anti-PD-1 immune checkpoint therapy and showed resistance to the treatment [199]. As previously discussed, follicular helper cells are known to facilitate the organization of TLSs, which can form due to chronic inflammation from autoimmune diseases, chronic infections, or cancer [200]. The TLS is associated with a favorable prognosis in various tumor types, including MIBC [148,201]. The expression of CXCL13 by follicular helper cells within the TLS is indicative of prolonged survival and enhanced clinical benefits from immune checkpoint inhibitors, such as pembrolizumab and atezolizumab [148,200].

Despite limited discussion in the literature regarding the distribution of chemokines across BCa subtypes, the immune cell distribution within each subtype is particularly relevant to chemokine signaling. Various research groups have delineated subtypes that further subclassify the classical luminal and basal categories, which may enhance the assessment of therapeutic efficacy for immunotherapies. The current consensus identifies six distinct subtypes within MIBC: luminal papillary, luminal non-specified, luminal unstable, stroma-rich, basal/squamous, and neuroendocrine-like. Additionally, three subtypes for NMIBC have been defined: class I, class II (a and b), and class III [202]. While the specific distribution of chemokines within these subtypes is seldom defined, their characteristics may contribute to elevated chemokine expression. For example, CAFs found in luminal non-specified and stroma-rich subtypes can secrete CXCL1, CXCL2, CXCL12, and CXCL14, contributing to tumor progression [202,203]. The distinction between the subtypes becomes relevant when considering the associated genetic mutations of each subtype. Specifically, PPARγ mutations are linked to both luminal non-specified and luminal unstable subtypes and are associated with immunosuppression, which may be reinforced by the production of TAM- and TAN-recruiting chemokines CXCL1 and CXCL2 [202,204]. This may help explain the overall poor prognosis associated with the luminal non-specified subtype compared to luminal unstable [202]. 

## 5. Therapeutic Targeting of Chemokines in Bladder Cancer

While the role of chemokines in cancer has been extensively studied, only three chemokine-targeting drugs have received FDA approval to date: maraviroc, which targets CCR5; plerixafor, which targets CXCR4; and mogamulizumab, which targets CCR4 [205,206,207]. Despite their approval for other indications, none of these drugs have yet been approved or tested in humans for BCa. However, preclinical studies have demonstrated their potential. For instance, mogamulizumab has shown promise in reducing tumor burden in a canine BCa-engrafted murine model by inhibiting CCL17-mediated recruitment of CCR4^+^ Tregs [208]. This finding suggests that targeting chemokine pathways could be a viable strategy for developing new therapies for BCa, highlighting the need for further research and clinical trials to explore their efficacy in this context (Table 2). 

### 5.1. Metformin Decreases Bladder Cancer Proliferation by Potentially NK Cell Infiltration and Cytotoxicity and Inducing Apoptosis

Metformin, a widely prescribed medication for type 2 diabetes, works by inhibiting hepatic gluconeogenesis and has been FDA-approved since 1994 [220]. However, recently, its role in malignancy has gained increasing attention. Specifically, in BCa, metformin has been found to inhibit the proliferation of BCa cells both in vitro and in vivo [211,212]. Metformin induced cell arrest in the BCa cell lines 5637 and T24 through the downregulation of cyclin D1, CDK1, and E2F1, along with the activation of AMP-activated protein kinase (AMPK) and the inhibition of mTOR [212]. The effect of metformin was also demonstrated in a humanized murine model, demonstrating the in vivo capacity of metformin [212]. Metformin induced downregulation of c-FLIP through the inhibition of mTOR/S6K1 signaling, which further enhanced TRAIL-mediated apoptosis in the TRAIL-sensitive BCa cell lines 253J and RT4 [211]. Jang et al. later showed that the inhibition of c-FLIP by metformin directly induces apoptosis through caspase cleavage and by translocation of apoptosis-inducing factor to the nucleus, independent of caspase cleavage [215]. Although not yet studied in BCa, metformin treatment was found to inhibit CXCL1 in head and neck squamous cell carcinoma (HNSCC) and esophageal squamous cell carcinoma (ESCC), suggesting a possible chemokine-mediated mechanism in metformin’s effect in BCa [213,214]. In ESCC, metformin-mediated activation of AMPK resulted in CXCL1 downregulation, leading to an overall decrease in PMN-MDSC infiltration while in HNSCC, metformin was found to inhibit CXCL1 and mTOR activation, resulting in increased NK cell infiltration and cytotoxicity [213]. 

### 5.2. Impact of CXCL8 Blockade on Tumor Burden in Bladder Cancer

Increased expression of CXCL8 by BCa cells has been shown to enhance the metastatic and angiogenic capacities of BCa through the regulation of MMP-2 and MMP-9 expression [221]. The efficacy of CXCL8 blockade in reducing bladder tumor burden has been demonstrated in vitro and in vivo using CXCL8-expressing BCa cell lines [222]. In vitro treatment of the BCa cell lines UM-UC3 and 253J B-V with the CXCL8 antibody ABX-IL8 resulted in decreased MMP-2/MMP-9 activity and reduced invasion [222]. Additionally, transfection of CXCL8 into the non-metastatic BCa cell line 253J-P increased MMP-9 expression as well as BCa invasion in vitro, which was abrogated following ABX-IL8 treatment [221,222]. Blockade of CXCL8 also reduced tumor burden in an orthotopic BCa model by downregulating NF-κB and subsequently lowering MMP-2 and MMP-9 expressions [222]. These findings suggest potential therapeutic efficacy in patients with CXCL8-high BCa, although CXCL8-producing immune infiltrates may also modulate this pathway. A Phase Ib clinical trial found HuMAX-IL8 (BMS-986253) to be safe and well tolerated in patients with advanced metastatic solid tumors (NCT02536469), while ongoing Phase 1b/2 trials are assessing its efficacy in combination with immunotherapy for hormone-sensitive prostate cancer (NCT03689699) and HNSCC (NCT04848116) [216,217,218]. However, HuMAX-IL8 has not been tested in BCa. 

### 5.3. ETV4 Regulates CXCL1/8 and TAN Migration in Bladder Cancer

Due to the redundancy of chemokine signaling and off-target binding of chemokine antagonists, treatments directly targeting chemokines and chemokine receptors are difficult to control once administered, evident by the lack of FDA-approved chemokine-targeting drugs. Decreasing the expression of elevated chemokines reduces signaling while avoiding these issues. ETV4 is a transcription factor upregulated in BCa and an upstream modulator of tumorigenesis-related gene transcription via the transcription of miRNAs [223]. ETV4 is also capable of directly binding to the promoter of both *CXCL1* and *CXCL8* following phosphorylation at Y392 [126]. The knockdown of ETV4 reduces CXCL8 and CXCL1 secretion in UM-UC3 and T24 cells, as does the mutation of tyrosine at Y392. This decrease subsequently impairs the migration of TANs and the CXCL1/CXCL8-induced production of VEGF-A and MMP-9 by TANs [126]. Targeting transcription factors such as ETV4 may be a promising therapeutic target for decreasing migration and invasion in BCa cells [126]. 

### 5.4. CXCL12/CXCR4 Interactions Activate Pro-Tumorigenic Pathways and Impact Wnt/β-Catenin and BCa Cell Proliferation

The stimulation of CXCR4 on BCa cells by CXCL12 enhances BCa cell migration and possibly the targeted metastasis to the lung, liver, bone marrow, and lymph nodes through gradient-dependent signaling [224]. Treatment of the TCCSUP BCa cell line with an anti-CXCR4 blocking antibody reduced migration and invasion in vitro [224]. Additionally, the CXCR4/CXCL12 pathway activates specific pro-tumorigenic pathways, including the Wnt/β-catenin pathway [219]. Suppression of CXCR4 with AMD3465 in the SW780 BCa cell line resulted in decreased expression of β-catenin, MMP-2, and c-Myc. Conversely, the addition of CXCL12 upregulated β-catenin and c-Myc expression, promoting cellular proliferation and metastasis. Inhibition of CXCR4 with AMD3465 in BCa cells significantly suppressed colony formation and growth. Xenograft studies confirmed that AMD3465 treatment significantly reduced bladder tumor weights compared to controls [219]. It is important to note that, in determining the expression of CXCR4 in BCa cell lines, the SW780 cell line, derived from a non-muscle-invasive bladder tumor, displayed higher transcriptional levels of CXCR4 and β-catenin than the 5637 and T24 BCa cell lines, both of which were derived from muscle-invasive bladder tumors [219,225]. Although the expression of CXCR4 in human bladder tumor samples correlates with tumor stage and is expressed highest in invasive bladder tumors, these results suggest a possible marker for the metastatic capacity of bladder tumors. Specifically, molecular analyses of BCa cell lines have shown that the SW-780 cell line has a missense mutation of the *FAT4* gene, a member of the cadherin superfamily and regulator of Wnt/β-catenin signaling [226]. Genetic analysis of bladder tumors, in addition to histopathology, may be used to provide a more cohesive picture of metastatic and proliferative capacities, as well as possible therapeutics, as opposed to histopathological tumor grading alone.

## 6. The Future of Chemokines in Bladder Cancer: Addressing Gaps and Exploring Future Directions

Chemokine research in BCa focuses broadly on whole-gene or -protein expression. However, post-translational modifications (PTMs), polymorphisms, and isoforms are often overlooked, leading to generalized or conflicting data on chemokine functions in BCa. 

Polymorphisms in CCL2 and CCR2 are well documented, with two functional polymorphisms of CCL2 investigated in BCa: rs1024611, an A→G mutation in the MCP-1 promoter region that upregulates CCL2 expression and rs3917887, an insertion/deletion in intron 1 of MCP-1 [227,228]. In a cohort of BCa in a North Indian population, rs3917887 was found to increase the risk of BCa, while rs1024611 was correlated with increased BCa risk in Caucasians and Mexican Mestizos [229,230]. Polymorphisms of the CCL2 receptor, specifically CCR2-V64I, have also been correlated with increased BCa risk [231]. Another polymorphism investigated in BCa is CXCL8 rs4073, an A→T mutation in the promoter region of the *CXCL8* gene [232]. In a North Indian cohort, the AA genotype of rs4073 was found to significantly correlate with increased BCa risk and a reduced risk of recurrence following BCG therapy [233]. The correlation, however, does not seem to be consistent in patients with other ethnic backgrounds [234]. Current BCa treatment regimens are generally based on tumor stage and muscle invasion; however, the ethnicity-specific effects are often overlooked. To efficiently target chemokines in BCa treatment, it is crucial to consider the isoforms of the chemokines present. However, disregarding the ethnicity-specific outcomes of these treatments could be detrimental to patients of specific ethnicities and require further research. 

PTMs occur after translation and are specific to proteins exposed to external factors that influence these modifications. PTMs result from enzymatic activity, such as that of peptidases and proteases, the addition of functional groups like methylation and acetylation, or chemical processes like deamination, oxidation, and nitration [235]. During inflammation, the upregulation and release of enzymes MMPs, plasmin, CD13, CD26, and peptidylarginine deiminases (PADs) and exposure to protein modifiers such as peroxynitrite, trigger the truncation, degradation, nitration, or citrullination of chemokines, fine-tuning their functions and influencing their immune cell recruitment profile. For example, a key modulator of PTM in cancer progression is CD26, or dipeptidyl peptidase 4 (DPP4), which is capable of cleaving proteins with an alanine or proline at the NH_2_ terminus [236]. In BCa, upregulation of DPP4 is correlated with aggressive and advanced-stage bladder tumors and the knockdown of DPP4 decreases the survival, proliferation, migration, and invasion of BCa cell lines [237]. While the connection between DPP4 and chemokines has not been explored in BCa, CXCL9, CXCL10, CXCL12, CCL22, CCL11, CCL5, and CCL7 are confirmed substrates of DPP4 [238,239,240]. The truncation of chemokines by DPP4 consistently results in reduced chemotactic potential, with truncations in CXCL10 resulting in antagonistic function [238,239,240]. With multiple FDA-approved DPP4 inhibitors available, exploring the role of DPP4 in BCa progression unlocks a new venue of commercially available drugs that can be used in BCa.

Elevated levels of MMPs, particularly MMP-2 and MMP-9, often correlate with increased chemokine levels [122,123,130,165], reflecting a complex interplay between these proteases and chemokine signaling in cancer. MMPs, a family of zinc-dependent endoproteinases, are crucial for degrading the extracellular matrix (ECM) and modulating cellular functions such as apoptosis and migration [241] They contribute to cancer progression and metastasis by facilitating tumor invasion, EMT, and angiogenesis through ECM degradation [242]. Additionally, MMPs target substrates beyond the ECM, including chemokines. Proteolytic cleavage of chemokines by MMPs can produce truncated proteins, which may either inactivate the chemokine or significantly alter its function, further complicating the chemokine signaling network in cancer. Their role in BCa remains uninvestigated, and future research should investigate the role of MMPs in BCa to better understand their impact on chemokine signaling and tumor progression.

In addition to truncation, other PTMs also impact chemokine functionality, such as nitration and oxidation. The cancer TME, particularly inflammatory cancers, is often rich in nitric acid and superoxide anions produced by neutrophils and macrophages [243]. Nitration of CCL2, CCL3, CCL5, CCL11, CXCL8, and CXCL12 has been shown in vitro, resulting in reduced or altered chemotactic capacity, with CCL2 nitration reducing its affinity to glycosaminoglycans, further diminishing signaling capacity [243]. Glycosylation of chemokines may also alter chemokine functionality, although the effect is not consistent between chemokines. Particularly, the glycosylation of CCL2 demonstrated reduced chemotaxis in vitro, while the glycosylation of CCL11 and CCL5 showed no effect [243]. Citrullination, a process mediated by PAD in which arginine is converted to citrulline, alters the charge of the protein [243]. Citrullination of CXCL5, CXCL8, and CXCL10 has been described to occur naturally, resulting in altered chemotactic capacity [243]. Despite their apparent redundancy, the chemokine network relies on tightly regulated PTMs to modulate activity and receptor affinity. These mechanisms must be thoroughly investigated in BCa before developing chemokine-targeted therapies to avoid paradoxical outcomes, which are common in chemokine research.

The role of atypical chemokine receptors (ACKRs) is also overlooked in BCa. The ACKR family is composed of ACKR1 (DARC), ACKR2 (D6), ACKR3 (CXCR7), and ACKR4 (CCX-CKR) [244]. Unlike typical chemokine receptors, ACKRs do not elicit G-protein signaling. Instead, they modulate immune signaling through mechanisms such as chemokine endocytosis and transcytosis, scavenging, and receptor internalization or desensitization via β-arrestin coupling [245]. The role of ACKRs in BCa remains largely unknown, despite their relevant expression in the disease. Specifically, ACKR3 expression is significantly elevated in BCa and is further increased in higher-grade tumors [246]. Increased ACKR3 expression on BCa cell lines has been found to enhance proliferation, migration, and invasion. This upregulation also elevates markers of EMT and metastasis in vivo, through the activation of Akt and ERK signaling pathways [246]. Additionally, ACKR3 binds to the pro-tumorigenic chemokine CXCL12. Elevated ACKR3 in BCa is correlated with increased expression of CXCL8 and VEGF [246]. Future research should focus on elucidating the specific roles and mechanisms of ACKRs in BCa, particularly ACKR3, to better understand their contribution to tumor progression and to explore potential therapeutic strategies targeting these receptors.

The prognostic value of chemokines in BCa is evident, yet the biochemical impact of the BCa TME on chemokines remains poorly understood. The structural integrity of chemokines is crucial for their functionality, as even minor changes significantly alter their chemotactic and signaling capabilities. Despite the highly mutagenic conditions within the TME, research on the structure of intratumoral chemokines is sparse. To develop novel drug treatments that effectively target chemokines in cancer, it remains crucial to first investigate the biochemical effects of the cancer environment on chemokine structures and stability.

## 7. Conclusions

With the increased focus on chemokine signaling in BCa, it is evident that the TME influences chemokine functionality in several ways. The unique inflammatory yet immunosuppressive environment of the TME, along with tumor-associated miRNAs and exosomes, can significantly impact chemokine behavior. Specifically, the TME may alter chemokine expression, modify chemokine activity through post-translational modifications, and affect chemokine interactions with receptors. Additionally, tumor-derived miRNAs and exosomes carry factors that further modify chemokine functions or their expression profiles. Investigating these interactions and mechanisms is crucial to delineate how chemokine functions are altered in BCa. Ultimately, a deeper insight into these chemokine axes may reveal novel therapeutic targets and enhance the efficacy of existing treatments, potentially leading to more effective and personalized therapeutic approaches for BCa. 

## Figures and Tables

**Figure 1 cancers-16-03303-f001:**
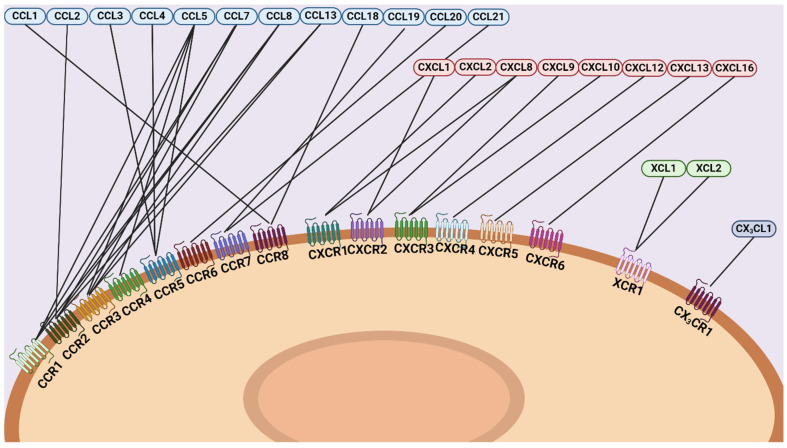
Chemokines implicated in cancer and their receptors. Shown is an illustration of the chemokine ligands and their respective receptors which have been implicated in cancer progression in humans, separated by subgroups (CC, CXC, CX_3_C, and XC). Original figure created with BioRender.com. https://app.biorender.com/illustrations/66b65196764066b50ca0c443 (accessed on 25 September 2024).

**Figure 2 cancers-16-03303-f002:**
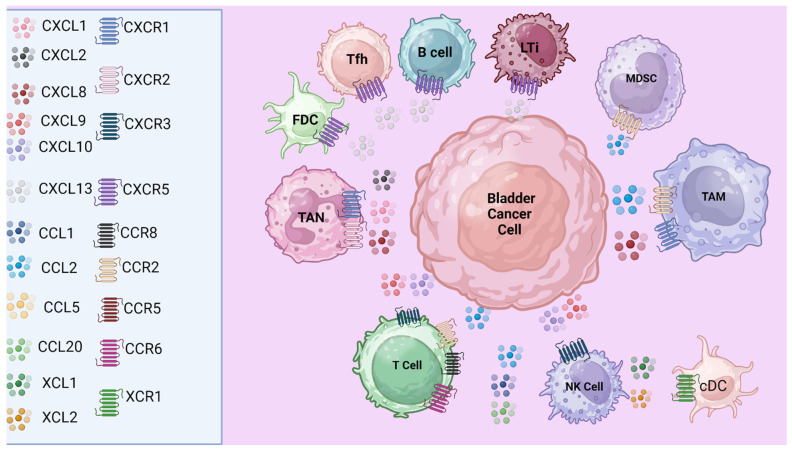
Chemokines mediate the recruitment of immune cells in BCa. The expression of chemokines by BCa cells can recruit both cytotoxic and immunosuppressive immune cells. TANs and TAMs expressing CXCR1 or CXCR2 are recruited by CXCL1, CXCL2, or CXCL8. Additionally, TAMs and MDSCs can be recruited through CCL2/CCR2 signaling, which also recruits T cells, in addition to CXCL9/CXCL10/CXCR3. The CXCL9/CXCL10/CXCR3 axis also recruits NK cells, which then recruit cDC through XCL1/XCL2/XCR1 and T cells through CCL20/CCR6, CCL1/CCR8, and CCL2/CCR2. The production of CXCL13 by stomal cells recruits LTis, B cells, Tfhs, and FDCs via CXCR5. FDC: follicular dendritic cell; Tfh: T follicular helper cell; LTi: lymphoid tissue-inducer cell; MDSC: myeloid-derived suppressor cell; TAM: tumor-associated macrophage; TAN: tumor-associated neutrophil; NK: natural killer cell; cDC: conventional type 1 dendritic cell. Original figure created with BioRender.com https://app.biorender.com/illustrations/663c045c26302c26221c2439 (accessed on 25 September 2024).

**Figure 3 cancers-16-03303-f003:**
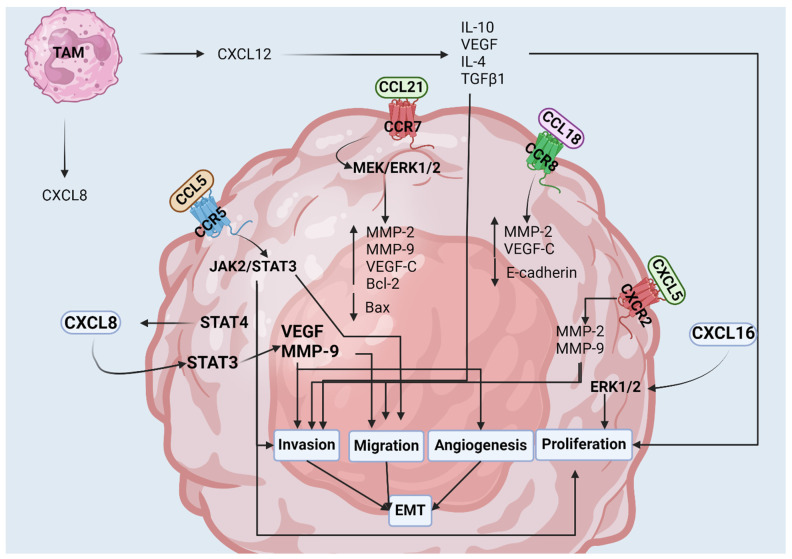
The expression of chemokine receptors on BCa cells activates pathways that promote tumorigenesis and metastasis. Chemokine signaling activates multiple pathways which modulate tumorigenesis and metastasis. Recruited immune cells, such as TAMs, also participate in the production of chemokines which further tumor progression. The production of MMPs and VEGF as a result of chemokine signaling most significantly contributes to EMT and metastasis due to their role in the loss of anchorage through ECM degradation and angiogenesis. TAM: tumor-associated macrophage; IL: interleukin; VEGF: vascular endothelial growth factor; TGFβ: transforming growth factor β; MMP: matrix metalloproteinase; EMT: epithelial–mesenchymal transition; ECM: extracellular matrix. Original figure created with BioRender.com. https://app.biorender.com/illustrations/66cbb4841d9552b0b40802ad (accessed on 25 September 2024).

**Table 1 cancers-16-03303-t001:** Chemokines and chemokine receptors and their role in BCa.

Chemokine Family	Chemokine Ligands	Associated Chemokine Receptor	Functions in BCa	References
CC Chemokines	CCL2	CCR2	Recruitment of TAMs, increases BCa cell migration and invasion.	[119]
	CCL5	CCR1, CCR3, CCR4, CCR5, ACKR1, ACKR2	Increases TAM recruitment, enhances the infiltration, invasion, and proliferation of BCa cells	[120,121]
	CCL18	CCR8	Induces migration, invasion, and EMT of BCa cells.	[122]
	CCL21	CCR7	Induces lymph node metastasis, increases proliferation of BCa cells.	[123,124]
CXC Chemokines	CXCL1	CXCR1, CXCR2	Recruitment of TAMs and TANs. TAN-derived MMP-9 and VEGF-A induce lymph node metastasis.	[125,126]
	CXCL2		Recruitment of TANs.	[125]
	CXCL8		Recruitment of TAMs and TANs. Increases the formation of NETs by recruited neutrophils. Increases survival, migration, invasion, angiogenesis, and lymph node metastasis of BCa.	[127,128,129,130]
	CXCL12	CXCR4, ACKR3	Recruitment of TAMs, increased proliferation and invasion of BCa	[131]
	CXCL13	CXCR5	Potent recruiter of CXCR5^+^ immune cells, induces the formation of tertiary lymphoid structures	[132]
	CXCL16	CXCR6	Elevated expression on tumor-infiltrating NK cells, contributes to T cell chemotaxis. Promotes BCa proliferation	[133,134]
CX_3_C Chemokines	CX_3_CL1	CX_3_CR1	Promotes the migration of BCa cells in vitro	[135]
XC Chemokines	XCL1	XCR1	Increased expression on tumor-infiltrating NK cells	[133]
	XCL2			

The interactions between chemokines and their respective receptors in BCa. BCa: bladder cancer; ACKR: atypical chemokine receptor; TAM: tumor-associated macrophage; TAN: tumor-associated neutrophil; MDSC: myeloid-derived suppressor cell; EMT: epithelial–mesenchymal transition; MMP-9: matrix metalloproteinase 9; VEGF-A: vascular endothelial growth factor A.

**Table 2 cancers-16-03303-t002:** Chemokine-targeting therapeutics in BCa.

Drug	Target	Current Use	
Mogamulizumab	CCR4	FDA-approved for the treatment of mycosis fungoides, current clinical trials only in cutaneous T-cell lymphomas and T-cell leukemias/lymphomas. Found to reduce tumor burden in a canine BCa xenograft murine model.	[208,209,210]
Metformin	CXCL1	Reduced bladder tumor burden in a humanized murine model. Inhibits CXCL1 in HNSCC and ESCC.	[211,212,213,214,215]
HuMAX-IL8	CXCL8	Phase Ib clinical trial in advanced metastatic solid tumors. Phase 1b/2 trials in combination with immunotherapy for hormone-sensitive prostate cancer and HNSCC.	[216,217,218] (NCT02536469) (NCT03689699) (NCT04848116)
AMD3465	CXCR4	Inhibits the in vitro proliferation, migration, invasion, and β-catenin expression in BCa cells. Reduced BCa cell growth in a murine xenograft model.	[219]

Current drugs currently undergoing clinical trials or approved for use in diseases other than BCa. AMD3465 has not been tested in humans due to the lack of oral bioavailability. BCa: bladder cancer; HNSCC: head and neck squamous cell carcinoma; ESCC: esophageal squamous cell carcinoma.

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
