# Peer review of "The Laws of Attraction: Chemokines as Critical Mediators in Cancer Progression and Immunotherapy Response in Bladder Cancer"

_cancers, 2024, doi:10.3390/cancers16193303_

Round 1

Reviewer 1 Report

Comments and Suggestions for Authors

The authors described main chemokines which play important roles in cancer progression, prognostic relevance, and candidate therapy targeting the chemokines in bladder cancer.

It is interesting and important themes for future treatment strategy in bladder cancer.

I have several questions and suggestions.

1. The indications of the individual therapies are important. For NMIBC (local reccurrence and invation)? For MIBC (metastasis)? For Metastatic lesions (progression)? I hope that you would mention the candidate indications of each therapy targeting the chemokines.

2. It is reported that invasive bladder cancer is categorized into intrinsic subtype  including "basal" and "luminal" by gene expressions. Are these chemokines related to the intrinsic subtype?

Reviewer 2 Report

Comments and Suggestions for Authors

   It is essential to revise the manuscript with a strong emphasis on distinguishing findings derived from human versus mouse data. This distinction is critical for drawing relevant conclusions that advance the translational potential of your research toward human clinical applications. Highlighting these differences will significantly enhance the manuscript's contribution to clinical care.

   The abstract should clearly articulate the central themes and scientific challenges that your manuscript addresses. This would sharpen the focus for readers and reviewers alike, allowing them to grasp the critical advances and implications of your findings from the outset. Consider explicitly stating how your findings address gaps in our understanding of chemokine dynamics in the tumor microenvironment (TME).

   While the authors have done well in detailing individual chemokine roles, it is crucial to discuss the interplay and balance between them. For instance, the contrasting roles of CXCL10 and CXCL12 within the tumor microenvironment should be explored in greater depth. This will provide a more nuanced understanding of how chemokine signaling modulates immune responses. I recommend referring to the following sources to support this discussion: 

   - NFκB-Activated COX2/PGE2/EP4 Axis Controls the Magnitude and Selectivity of BCG-Induced Inflammation in Human Bladder Cancer Tissues. Cancers (Basel). 2021 Mar 16;13(6):1323. doi: 10.3390/cancers13061323. PMID: 33809455; PMCID: PMC7998891. 

   - Breaking Barriers: Modulation of Tumor Microenvironment to Enhance Bacillus Calmette-Guérin Immunotherapy of Bladder Cancer. Cells. 2024 Apr 18;13(8):699. doi: 10.3390/cells13080699. PMID: 38667314; PMCID: PMC11049012.

   A thorough discussion is needed on how immunosuppressive chemokines contribute to the development of immune exhaustion markers. Connect these observations to the clinical use of immune checkpoint inhibitors, providing insights on how chemokine modulation could enhance therapeutic outcomes. A more integrated clinical perspective here would strengthen the manuscript’s relevance for translational applications.

   Although the role of chemokines in the bladder cancer epithelium has been sufficiently discussed, it is equally important to highlight their role in shaping the myeloid cell-rich "shield" that often protects the tumor from immune clearance. Addressing how this shield contributes to the immunosuppressive nature of the TME will provide a more complete understanding of chemokine function in the tumor’s defense against immunotherapy.
